# Assessment of the Bearing Capacity of Bridge Foundation on Rock Masses

Ana Alencar [1], Rubén Galindo [1,*], Claudio Olalla Marañón [1] and Svetlana Melentijevic [2]

[1] Departamento de Ingeniería y Morfología del Terreno, Universidad Politécnica de Madrid, 28040 Madrid, Spain; at.santos@alumnos.upm.es (A.A.); claudio.olalla@upm.es (C.O.M.)
[2] Departamento de Geodinámica, Estratigrafía y Paleontología, Universidad Complutense de Madrid, 28040 Madrid, Spain; svmelent@ucm.es
* Correspondence: rubenangel.galindo@upm.es

**Abstract:** This paper aims to study the bearing capacity of a shallow foundation on rock mass, considering the most usual bridge footing width and adopting a Hoek–Brown material. The dimension of the foundation has been shown to be very significant in soils with linear failure criteria (Mohr–Coulomb envelope), and its study is necessary in the case of non-linear failure criteria, typical of rock masses. Analytical solutions do not allow incorporating this effect. A parametric study by a finite difference method was carried out, studying a wide variety of rock mass through sensitivity analysis of three geotechnical parameters: geological origin of the rock mass ($m_i$), uniaxial compressive strength, and geological strength index. The results obtained by the numerical solution for the Hoek–Brown failure criterion were compared with the analytical results by adopting the classical hypotheses of plane strain conditions, associated flow rule, and weightless rock mass. The variation of the numerical bearing capacity due to the consideration of the self-weight of the rock mass was also analyzed since its influence is conditioned by the volume of ground mobilized and therefore by the width of the foundation. Considering the similarities observed between the numerical and analytical results, a correlation factor function of the self-weight is proposed. It can be used in conjunction with the analytical method, to estimate in a semi-analytical way the bearing capacity of a bridge foundation.

**Keywords:** bearing capacity; finite difference method; bridge foundation; shallow foundation; Hoek–Brown failure criterion

## 1. Introduction

Over the years, several methods were used to study bearing capacity: limit equilibrium method [1,2], slip line method [3], limit analysis method [4,5], numerical methods of finite elements (FEM) or DLO [6–8], and artificial intelligence techniques [9].

The traditional analytical solutions to estimate the ultimate bearing capacity in soils [1,10] were developed for the linear Mohr–Coulomb failure criterion that depends on the cohesion and internal friction angle of the material, using for rock masses the equivalent strength parameters [11–13]. The non-linear Hoek–Brown failure criterion [13] is applied to rock masses when by inexistence or by abundance of discontinuities they have the same physical properties in all directions.

In the past 30 years, some analytical methods were developed to estimate the ultimate bearing capacity of rock mass based on the parameters that define the failure criterion. The analytical method that solves the internal equilibrium equations combined with the failure criterion was proposed by Serrano and Olalla [14] and Serrano et al. [15–18] by applying the Hoek–Brown [11] and the modified Hoek–Brown failure criterion [12], respectively. Furthermore, this theoretical method has been applied to more current failure criteria for general rock masses [19] and volcanic rocks [20]. The method was based on the characteristic line method [3] whose solution is independent of the foundation width

and therefore implies the hypothesis of weightless rock. In the case of rock mass, it was expected that the influence of self-weight becomes more significant for low-quality rocks and when the foundation dimensions become important given the large amount of the rock involved, that is the case of many bridge foundations. Therefore, and as indicated by Bower [21], the characteristic line method can be used to calculate the analytical solution of solids idealized in terms of the plastic failure criterion; that is perfectly rigid without considering the width of the foundation.

The method proposed by Serrano et al. [15] applies the theory of characteristic lines, by adopting the plane strain hypothesis, the associated flow rule, the coaxiality, the perfectly plastic yield surface, and the weightless rock mass (therefore independent of width). In addition, the slope at the edge of the foundation with inclination can also be analyzed with this method. This analytical formulation of the ultimate bearing capacity introduced a bearing capacity factor which makes the failure load proportional to the uniaxial compressive strength of the rock (*UCS*), as described in Section 2 below.

The similar structure of the equation that relates the ultimate bearing capacity to the uniaxial compressive strength is observed in other formulations, such as [22], recommended by AASHTO [23], and based on the lower bound solution adopting the hypothesis of not considering the width of the foundation.

The Eurocode 7 [24] includes a rather simplified method, in which, as a function of the rock type, the ultimate bearing capacity is estimated depending on the uniaxial compressive strength and the spacing of the main joint set. According to Miranda et al. [25], the method proposed in the Eurocode 7 [24] is rather simplistic and does not take into account important aspects that influence the bearing capacity like the depth of the foundation, its shape, the eccentricity of the load, the presence of water, etc.

The use of numerical methods and the progress of computational geomechanics have allowed access to the practical calculation of more sophisticated bearing capacity problems involving: (a) anisotropic rock masses (in [26] a solution in some simple cases is indicated and in [27] another solution for piles in rock masses is developed but in both cases the solutions do not depend on the width); (b) the presence of a water table in the rock [28], the shape of the foundation in rock masses [29], and roughness of the rock [30] where the results do not incorporate the size foundation; (c) the interaction with other structural elements such as tunnels [31] where it is not possible to know the influence of the dimension of the foundation; (d) bilayer rock under the footing [32] that does not depend on the width of footing; (e) the dynamic response of the foundation that offer solutions that do not depend on the geometric characteristics of the analyzed foundation (in [33] and [34] cases without self-weight of the ground are considered, [35] a particular application is considered but it is not possible to obtain the influence of the foundation size). In addition, new calculation methods have been developed in the study of piloted foundations in non-cohesive soil [36], cohesive soil [37], and considering inclined load [38] but using a linear failure criterion not representative of the behavior of a rock masses.

Different calculations using the finite element method under lower and upper bound theorem, developed, respectively, by Sloan [4] and Sloan and Kleeman [5], were used by Zheng et al. [39] and Sutcliffe et al. [40] to determine the bearing capacity of the fractured rock and jointed rock mass. Later, Merifield et al. [7] applied the limit theorems (upper and lower bound), as an extension of the formulation developed by Lyamin and Sloan [41,42] to determine the bearing capacity on a fractured rock mass whose behavior is a Hoek–Brown type. Merifield et al. [7] observed that the use of Mohr–Coulomb equivalent strength parameters overestimates the bearing capacity by up to 157% in the case of a good quality rock mass (*GSI*, Geological Strength Index, about 75). They also concluded that the Serrano et al. method [15] was the closest to the numerical results when the width of the foundation is not considered, although with $GSI \leq 10$ this method is very conservative and underestimates the bearing capacity factor. In addition, the Carter and Kulhawy method [20], however, is rather conservative and is typically 30–80% below the average finite element results. Furthermore, it has been observed that the influence of the

foundation width on the bearing capacity decreases as the value of the *GSI* and the *UCS* increases, thus, concluding that the dimension of foundation should be always considered for low quality rock mass with *GSI* less than approximately 30.

Yang and Yin [43] applied two techniques to calculate the bearing capacity: (1) the multi-wedge translation failure mechanism, that can take into account the foundation width and the external surcharge; and (2) the tangential line technique, that was originally used to analyze the slope stability, where the same effects of the foundation width and the surcharge on the bearing capacity were not considered. They used the upper bound limit theory for strip foundation based on a modified Hoek–Brown failure criterion deducing the equivalent parameters of the Mohr–Coulomb failure criterion. Saada et al. [44] also proposed another method to calculate the bearing capacity based on the limit theories by applying the Hoek–Brown failure criterion and deducing the equivalent parameters of the Mohr–Coulomb method that provided a better fit than the results obtained by [43]. These methods show the importance of the dimension of the foundation in the bearing capacity, however, their implementation requires a specific analysis for each case.

Keshavarz and Kumar [45] and Galindo et al. [46] undertook an evaluation of the bearing capacity using the method of characteristics lines with pseudostatic load but without the consideration of the foundation width.

Tajeri et al. [47] and Alavi and Sadrossadat [48] applied the linear genetic programming (LGP) models to estimate the bearing capacity of shallow foundations on rock masses, using a database with 102 experimental data sources from different studies. This work provides an adequate calculation method when studying foundations on rock masses whose failure mechanism is induced by the local fracturing conditions under the foundation, since the data come from load tests with reduced dimensions; however, it is not suitable for use in conditions of global failure, which is very common in large foundations, typical of bridges.

Existing analytical solutions [15] allow particular configurations and cannot consider the influence of the foundation dimension. Its consideration can be introduced as a corrective factor to the theoretical formulation to generate a semi-empirical formulation. This process was carried out in soil mechanics from Brinch–Hansen solutions widely used today, where an empirical factor that depends on the density of the ground and is proportional to the width of the foundation is incorporated. In Rock Mechanics, however, there are no corrective factors in the literature that allow us to introduce the effect of the weight of the ground affected by the width of the foundation on the analytical formulation. Thus, this research allows to obtain this correction factor ($W_F$) by numerical experimentation and offers a complete semi-empirical formulation that considers the dimension of the foundation in the analytical formulation of the bearing capacity of the rock masses. Therefore, it is possible to estimate a bridge foundation bearing capacity with more accuracy, once the foundation dimension is considered.

## 2. Analytical Formulation for the Ultimate Bearing Capacity

As is generally known, in rock mechanics, the non-linear Hoek–Brown failure criterion [13] is the most used and it is applicable for the rock mass and is formulated in function of the major principal stress ($\sigma_1$) and minor principal stress ($\sigma_3$) according to the following equation:

$$\frac{\sigma_1 - \sigma_3}{\sigma_c} = \left( m \cdot \frac{\sigma_3}{\sigma_c} + s \right)^a \tag{1}$$

The uniaxial compressive strength (*UCS*) is $\sigma_c$, while the parameters *m*, *s*, and *a* can be evaluated following [13] by Equations (2)–(4) and depend on the rock type ($m_i$), geotechnical quality index of the rock mass (*GSI*), and damage in the rock mass due to human actions (*D*) that in shallow foundations is usual equal to zero.

$$m = m_i \cdot e^{\frac{GSI - 100}{28 - 14 \cdot D}} \tag{2}$$

$$s = e^{\frac{GSI - 100}{9 - 3 \cdot D}} \tag{3}$$

$$a = \frac{1}{2} + \frac{1}{6} \cdot \left( e^{\frac{-GSI}{15}} - e^{\frac{-20}{3}} \right) \tag{4}$$

Serrano et al. [15] proposed an analytical formulation for estimating the ultimate bearing capacity of the shallow foundation independently of width, based on the modified Hoek–Brown failure criterion [13], taking into account the associated plastic flow rule, and strain plane.

A brief summary of the analytical formulation is presented below, since it allows obtaining a closed solution to the result, for which in this research it is intended to incorporate a corrective coefficient that directly allows considering the effect of the width of the foundation.

According to this analytical formulation the ground surface that supports the foundation is composed of two sectors (Figure 1): boundary 1 (free) with the inclination $\alpha$ where the load acting on a surface $f_1$ acting with the inclination of $i_1$ is known (for example, the self-weight load on the foundation level or the load from installed anchors); and the boundary 2 (foundation), where the bearing capacity of the foundation $P_h$ should be determined (acting with the inclination of $i_2$).

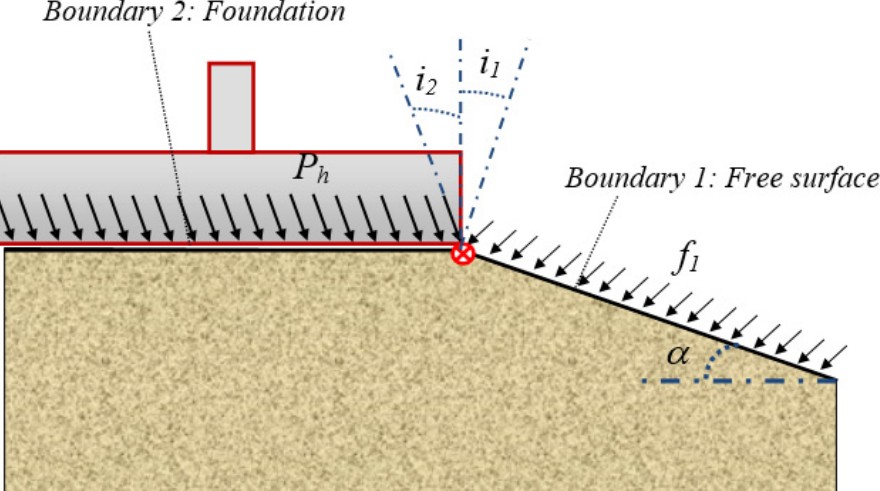

**Figure 1.** Mathematical model of the bearing capacity of the strip footing.

The analytical solution [15] based on the characteristic line method needs the equation of the Riemann invariants ($I_a$) fulfilled along the characteristic line:

$$I_a(\rho_1) + \psi_1 = I_a(\rho_2) + \psi_2 \tag{5}$$

$$I_a(\rho) = \frac{1}{2 \cdot k} \cdot \left[ \cot(\rho) + \ln\left( \cot\left(\frac{\rho}{2}\right) \right) \right] \tag{6}$$

In this equation, the instantaneous friction angle at the boundary 2 ($\rho_2$) is the only unknown, because the other variables can be defined at the boundary 1: instantaneous friction angle at the boundary 1 ($\rho_1$) and the direction of the principal stress in this sector ($\Psi_1$), thus expressing $\Psi_2$ in function of $\rho_2$. Knowing $\rho_2$ the ultimate bearing capacity can be estimated.

Through the analytical method [15] the bearing capacity is obtained by (7).

$$P_h = \beta_a \cdot \left( N_\beta - \zeta_a \right) \tag{7}$$

The resistant parameters $\beta_a$ and $\zeta_a$ defined by Serrano et al. [15] are applied to make dimensionless the calculation of the Hoek–Brown failure criterion. $\beta_a$ represents the characteristic strength that has the same units as *UCS* and is used to make pressures

dimensionless, while $\zeta_a$ ("tenacity coefficient") is a dimensionless coefficient that multiplied by $\beta_a$ corresponds to the tensile strength.

$$\beta_a = A_a \cdot UCS; \; \zeta_a = \frac{s}{(m \cdot A_a)}; \; A_a = \left( \frac{m \cdot (1-a)}{2^{\frac{1}{a}}} \right)^{\frac{1}{k}}; \; k = \frac{(1-a)}{a} \tag{8}$$

$A_a$, $k$ and the exponent $a$, are constants for the rock mass, and depend on rock type ($m$), *UCS* and *GSI*.

$N_\beta$ is the bearing capacity factor and it can be calculated, according to the problem statement, as follows.

The angle of internal friction $N_\beta$ can be obtained by iteration from the load at the boundary 1. From the value of $\rho_1$ and by the iteration of (5) the value of the internal friction angle at the boundary 2 ($\rho_2$) can be calculated. Finally, using $\rho_2$, the bearing capacity factor ($N_\beta$) proposed by Serrano et al. [15] can be calculated:

$$N_\beta = cosi_2 \left( \frac{1 - sin\rho_2}{k \, sin\rho_2} \right)^{\frac{1}{k}} \left( \frac{a(1 + sin\rho_2)}{sin\rho_2} cosi_2 + \sqrt{1 - \left[ \frac{a(1 + k \, sin\rho_2)}{sin\rho_2} sini_2 \right]^2} \right) \tag{9}$$

## 3. Numerical Analysis

This section describes the numerical method used to solve the study problem, including the hypotheses used and the calculation cases carried out. Calculations were developed to model accurately the bearing capacity of real bridge foundations and to compare to the described analytical solution. To do so, numerical simulations were run using FLAC software (Itasca, Version 7) [49], which uses an explicit finite difference (FDM) formulation.

The hypotheses incorporated in the numerical models are in accordance with those of the analytical method presented in the previous section since they will allow to compare these solutions and obtain numerically a correction factor to estimate the effect of the width of the foundation in the analytical solution. These hypotheses are as follows: the associated flow rule, plane strain condition, and without the self-weight of the rock. The numerical model was adopted in a way that the vertical load was directly applied on the ground surface (nodes), so that the characteristics of the foundation and the interaction with the ground surface did not influence the result. A symmetrical model was used, where only half of the strip footing was represented; and the boundaries of the model were located at a distance that did not interfere with the result (see Figure 2); in particular, it has been verified for all the calculated models that the reactions obtained in the boundaries are negligible. The modified Hoek–Brown constitutive model is incorporated into FLAC v.7 using the values of the parameters indicated in the Table 1, where a wide variety of types and states of rock masses are covered, considering a range of usual values in bridge foundations: 4 possible widths are entered, 4 possible values of *UCS* (values chosen for the rocks that determine the rock-foundation study, with respect to the *UCS* of the concrete), another 4 of the rock type parameter $m_i$, and 3 different values of *GSI*. Therefore: $4 \times 4 \times 4 \times 3 = 192$ different models are solved.

**Table 1.** Summary of the adopted parameters.

| $m_i$ | B (m) | UCS (MPa) | GSI |
|:---:|:---:|:---:|:---:|
| 5 (claystone) | 4.5 | 5 | |
| 12 (gypsum) | 11 | 10 | 10 |
| 20 (sandstone) | 16.5 | 50 | 50 |
| 32 (granite) | 22 | 100 | 85 |

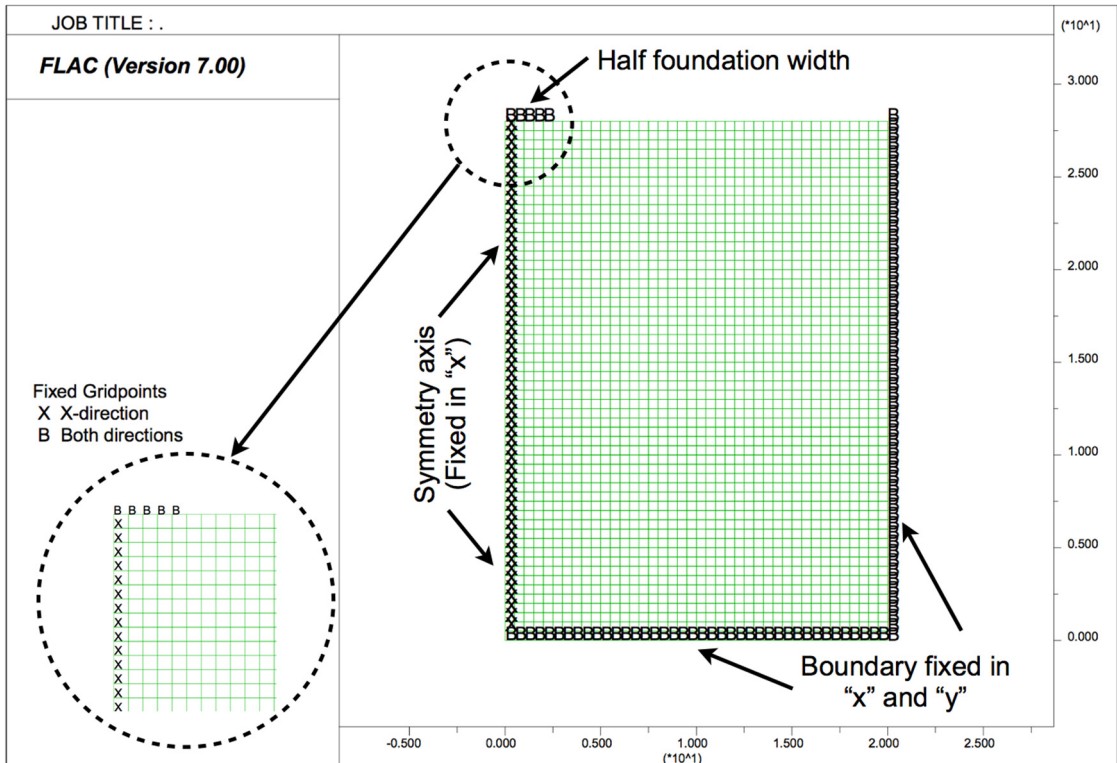

**Figure 2.** The 2D model employed in the numerical calculation (*x*–axis and *y*–axis units in meters).

Later, the load due to the self-weight of the rock mass (unit weight of 26 kN/m$^3$) was incorporated in the numerical calculations to know the influence of considering the foundation dimension. Therefore, it is necessary to incorporate the effect of the self-weight of the ground, typical of slope stability analysis in rock [50,51], in order to study the influence of the foundation geometry.

In the numerical analysis it was assumed that the bearing capacity was reached when the continuous medium did not stand more load, because an internal failure mechanism had formed. The load in FLAC was applied through velocity increments on the nodes that must be previously fixed, and the bearing capacity was known from the relation between stresses and displacements of one of the nodes: for this study we considered the central node of the foundation.

In order to compare the results obtained numerically (FDM) with the analytical solution [15], the same boundary conditions were used in the rock–foundation contact. However, the hypothesis of a rigid or flexible foundation depends on the relative deformability of the shallow foundation regarding the supporting ground. Therefore, this consideration cannot be included in the analytical solution since the bearing capacity is independent of the ground deformability. The conditions of uniform stresses and displacements at the contact points cannot be simultaneously imposed, and therefore, it is necessary to previously investigate how to consider the rigidity of the foundation using the FDM method. The best approximation of the numerical result with the analytical solution was observed assigning a constant deformation velocity for the contacting points of the foundation and the rock.

A convergence study was carried out controlling the value of the bearing capacity obtained under different increments of the applied velocity. Figure 3 shows the dependence of the results of the bearing capacity in relation to the velocity increments applied on the nodes, and how, with the decrease in the value of velocity increments, the result converges towards the final value. This convergence study was carried out for each case with a different combination of geometrical and geotechnical parameters.

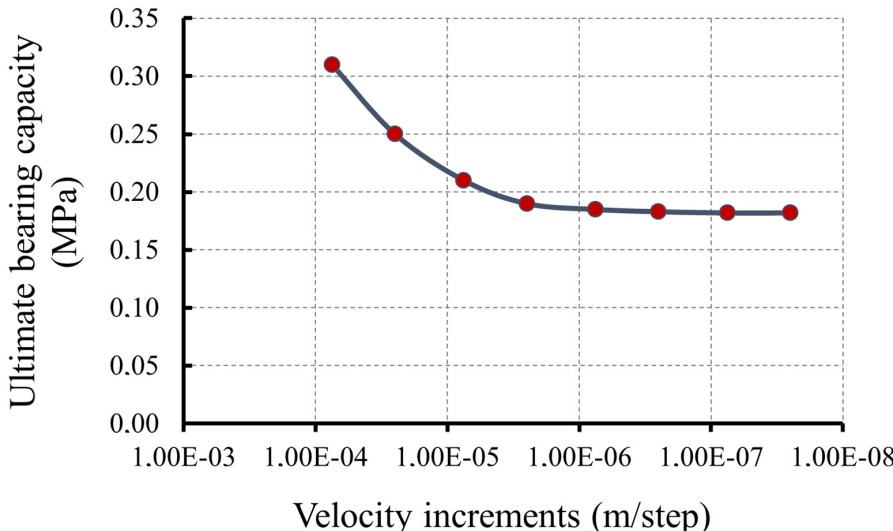

**Figure 3.** The bearing capacity vs. velocity increments diagram of the central node of the footing ($m_i$ = 5, $B$ = 4.5 m, *UCS* = 5 MPa, and *GSI* = 10).

## 4. Results and Discussion

### 4.1. Bearing Capacity for Weightless Rock

We studied the comparison of the numerical and analytical results on rock mass through a sensitivity analysis, where the influence of four geotechnical and geometrical parameters ($m_i$, *UCS*, *GSI*, and *B*) is observed. In Figure **1**, the correlation between the numerical ($P_{hFDM}$) and the analytical ($P_{hS\&O}$) results can be observed, thus concluding that the value of $P_{hFDM}$ is always higher than $P_{hS\&O}$, with a variation between the results of up to 60%.

Figure 4 shows that the two parameters that mostly influenced the relation between $P_{hFDM}$ and $P_{hS\&O}$ were $m_i$ and *GSI*, because the dispersion range changes considerably in function of the $m_i$ and *GSI* value (represented in the abscissa axis). Mathematical correlations will be indicated in the next section for a more detailed analysis.

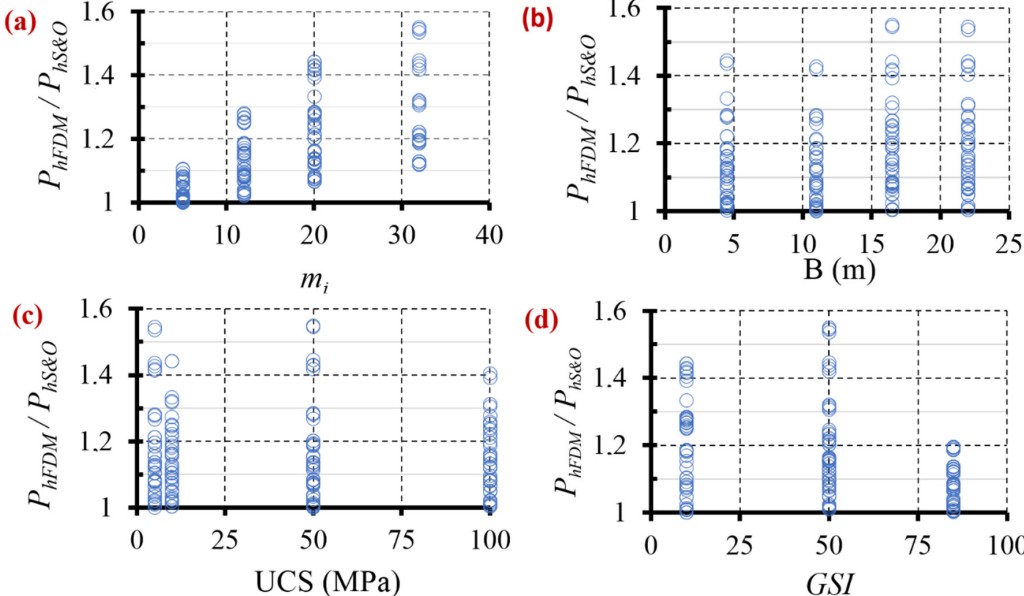

**Figure 4.** Correlation of $P_{hFDM}$ and $P_{hS\&O}$ in function of: (**a**) $m_i$; (**b**) *B*; (**c**) *UCS*; (**d**) *GSI*.

Figure 4a clearly demonstrates the influence of the rock type ($m_i$) on the correlation between $P_{hFDM}$ and $P_{hS\&O}$, with the results for adjusted greater values of $m_i$ being worse.

The results for $P_{hFDM}$ in the cases with lower $m_i$ ($m_i = 5$) were very similar to the results for $P_{hS\&O}$ presenting a variation lower than 15%, while for higher values of $m_i$ ($m_i = 32$) the numerical results ($P_{hFDM}$) could exceed up to 60% of the analytical results ($P_{hS\&O}$).

Figure 4b does not show a clear influence of $B$ on the relation between $P_{hFDM}$ and $P_{hS\&O}$, by increasing the dispersion ratio with the enlargement of $B$. This sensitivity influence is clearer in Figure 5 where the correlation of the results between $P_{hFDM}$ and $P_{hS\&O}$ is presented as a function of $P_{hS\&O}$. For a large footing ($B = 22$ m), the values of $P_{hFDM}$ are slightly above the values obtained for the smaller footing ($B = 4.5$ m). However, there is hardly any variation between widths of 16 and 22 m, or between 4.5 and 11.5, respectively. In other words, the influence of the foundation width has little influence (less than 15%) due to the non-incorporation of self-weight.

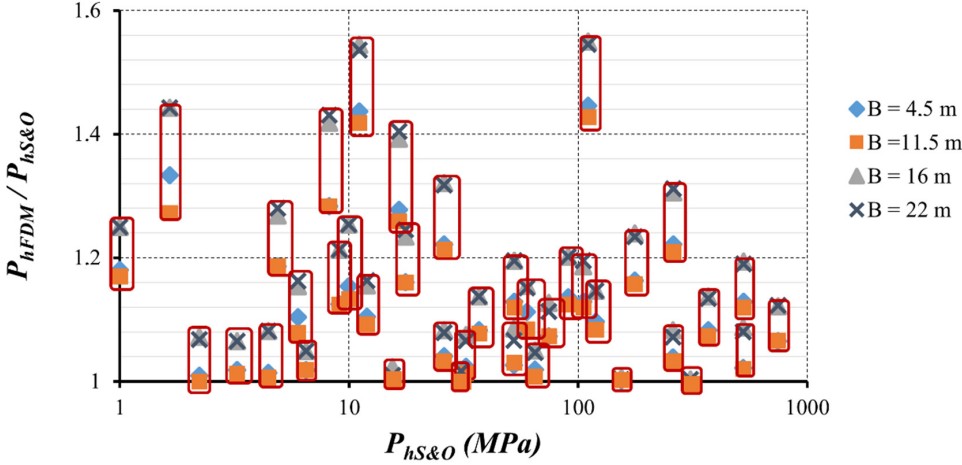

**Figure 5.** Relation between $P_{hFDM}$ and $P_{hS\&O}$ depending on $P_{hS\&O}$ and $B$.

Regarding the *UCS* parameter, Figure 4c shows that this parameter does not influence the correlation of $P_{hFDM}$ and $P_{hS\&O}$. The cause of this fact is that the numerical and analytical solution is proportional to the value of *UCS*; that is, the relationship $P_{hFDM}/P_{hS\&O}$ (and therefore the deviation of results) is independent of *UCS*.

Table 2 shows two examples of the correlation of $P_{hFDM}$ and $P_{hS\&O}$ with the variation of *UCS*. The first group of case studies (1, 2, 3, and 4) correspond to a rock type of claystone with $m_i = 5$ and a similarity in the results of $P_{hFDM}$ and $P_{hS\&O}$ can be observed. The second group of case studies (5, 6, 7, and 8) have sandstone with $m_i = 20$, the results of $P_{hFDM}$ and $P_{hS\&O}$ present a dispersion close to 25%. Based on these two examples, it is observed that a higher value of $m_i$ implies a greater dispersion between $P_{hFDM}$ and $P_{hS\&O}$ and the same dependence of the value of $P_h$ with the *UCS*.

**Table 2.** Results of $P_{hFDM}$ and $P_{hS\&O}$ for different case studies.

| Cases | | *UCS* (MPa) | $P_{hFDM}$ (MPa) | $P_{hS\&O}$ (MPa) |
|---|---|---|---|---|
| $m_i = 5$<br>$B = 11$ m<br>$GSI = 10$ | 1 | 5 | 0.22 | 0.22 |
| | 2 | 10 | 0.46 | 0.44 |
| | 3 | 50 | 2.22 | 2.22 |
| | 4 | 100 | 4.47 | 4.44 |
| $m_i = 20$<br>$B = 11$ m<br>$GSI = 10$ | 5 | 5 | 1.05 | 0.82 |
| | 6 | 10 | 2.1 | 1.65 |
| | 7 | 50 | 10.5 | 8.18 |
| | 8 | 100 | 20.8 | 16.52 |

In contrast, Figure 4d shows that the dispersion and the difference between the results decrease with the increase of *GSI*. With higher values of *GSI*, the dispersion range between

the results is less than 20%. For *GSI* = 10 the dispersion ratio is lower than 50%, while for *GSI* = 50 the difference is much greater.

Figure 6 represents the relation between $P_{hFDM}$ and $P_{hS\&O}$ depending on $P_{hS\&O}/UCS$ as a function of the *GSI*. This figure represents the relationship of bearing capacity results by eliminating the *UCS* parameter, which, as indicated, affects both methods in the same way. Thus, it can be concluded that a higher value of the bearing capacity implies a lower dispersion of results between both solutions.

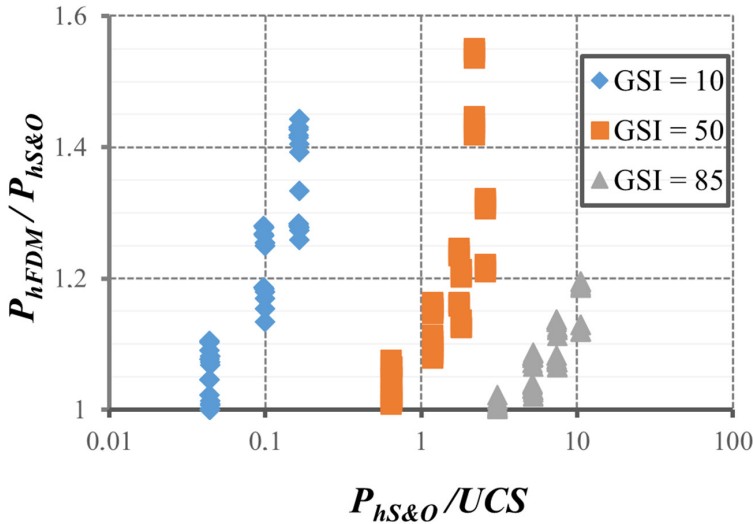

**Figure 6.** Relation between $P_{hFDM}$ and $P_{hS\&O}$ depending on $P_{hS\&O}/UCS$ in function of the *GSI*.

The other parameters that were studied ($m_i$, *B*, *UCS*) also had an influence on the value of the bearing capacity, but they did not clearly define the variation range of the bearing capacity as the value of *GSI* (Figure 6).

As mentioned above, we can conclude that the correlation of the results mainly depends on the combination of the values of *GSI* and $m_i$.

Figure 7 shows that the range of dispersion depends on the combination of the *GSI* with the $m_i$, varying from 10% for lower $m_i$ to more than 60% for a larger $m_i$. Therefore, the increase in the value of $m_i$ implies the increase of the dispersion range in the results. Qualitatively, it is also observed that when *GSI* is greater and $m_i$ is lower, the analytical and numerical results are very similar with a dispersion of less than 5%.

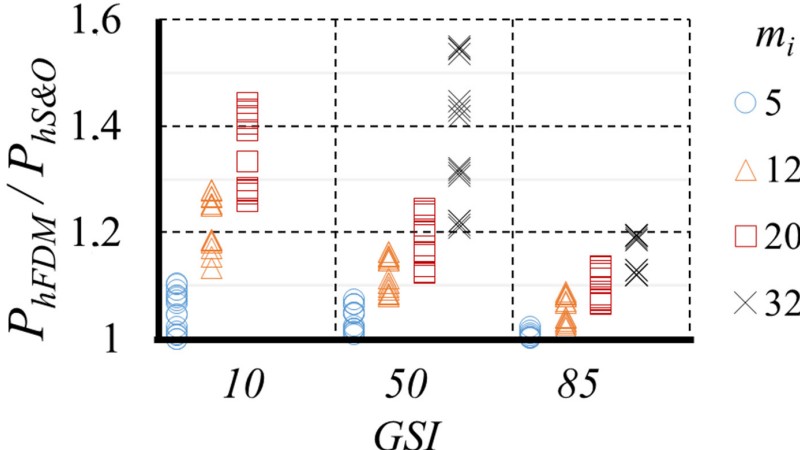

**Figure 7.** Relation between $P_{hFDM}$ and $P_{hS\&O}$ depending on *GSI* and $m_i$.

### 4.1.1. The Correlation between Numerical ($P_{h_{FDM}}$) and Analytical ($P_{h_{S\&O}}$) Results

Once the most determining parameters are known, it is possible to develop a correlation coefficient to estimate the percentage of variation between $P_{h_{S\&O}}$ and $P_{h_{FDM}}$, depending on different parameters of the rock mass.

We observed that the numerical results were always greater than the analytical results, so that the correlation can be expressed:

$$P_{h_{FDM}} = P_{h_{S\&O}} + \Delta P_h \tag{10}$$

To make (10) dimensionless, everything is divided by $P_{h_{S\&O}}$ and therefore in this research:

$$\frac{P_{h_{FDM}}}{P_{h_{S\&O}}} = 1 + \frac{\Delta P_h}{P_{h_{S\&O}}} \tag{11}$$

Knowing that the most influential parameters are *GSI* and $m_i$, the correlation of the results for each value of the *GSI* can be analyzed according to the rock type (Figure 8) using three equations with acceptable correlation coefficient. In Table 3 these equations are summarized.

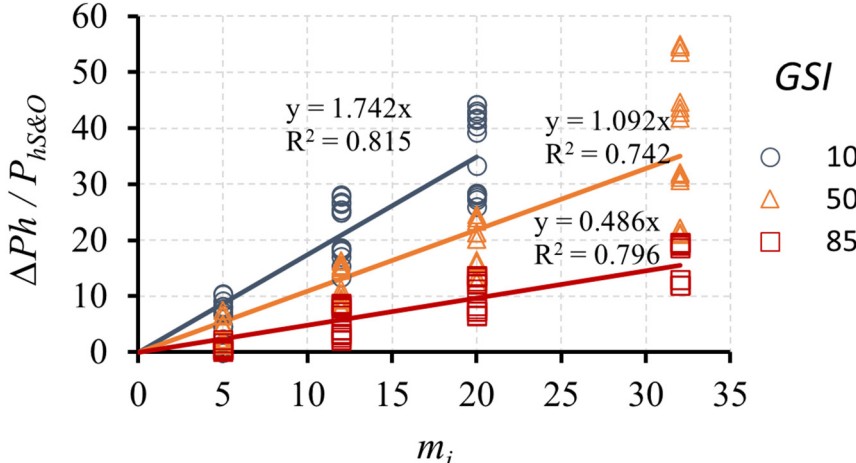

**Figure 8.** Relation between $\Delta P_h/P_{h_{S\&O}}$ depending on $m_i$, for different values of *GSI*.

**Table 3.** Equations $\Delta P_h/P_{h_{S\&O}}$ depending on $m_i$, for different values of *GSI*.

| GSI | Equations |
|---|---|
| 10 | $\frac{\Delta P_h}{P_{h_{S\&O}}} = 1.742 \cdot m_i$ |
| 50 | $\frac{\Delta P_h}{P_{h_{S\&O}}} = 1.092 \cdot m_i$ |
| 85 | $\frac{\Delta P_h}{P_{h_{S\&O}}} = 0.486 \cdot m_i$ |

Therefore, because the equations in Table 3 have the same structure, a single equation based on *GSI* is formulated in this research:

$$\frac{\Delta P_h}{P_{h_{S\&O}}} = \frac{100 - GSI}{50} \cdot m_i \tag{12}$$

Figure 9 shows the correlation between analytical result and the numerical solution (divided by the coefficient $(1 + \Delta P_h)$ to compare results without self-weight). From this figure, it can be concluded that the percentage variation calculated by (Equation (12)) had a good fit in the 192 cases that were studied, with a variation between the results that did not exceed 4%, emphasizing that most of the results were concentrated in the initial area of the graph.

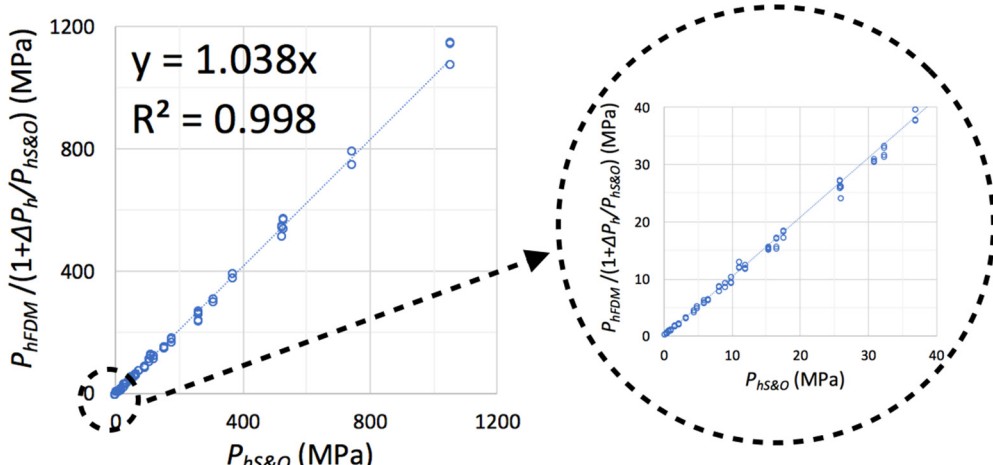

**Figure 9.** Correlation between numerical and analytical results without self-weight application (12).

### 4.1.2. Displacement Analysis

The two calculation methods used in this study are based on hypotheses that do not coincide in all aspects, therefore, it is expected that the results obtained are similar, but not the same.

In the numerical calculation, a stress path is formed until the failure is reached, while in the analytical method the equivalent failure stress is studied directly, taking into account the whole wedge of the ground below the footing. Therefore, in the numerical calculation, there can be other types of failure, such as the localized failure at the edge of the footing, which is not considered in the analytical solution.

The graphic output of the displacements (horizontal and vertical) developed below the foundation using FDM is presented in this section to understand how the failure mechanisms affect the results.

In the cases studied for this research, the local failure is not observed, since all the numerical results exceeded the analytical results and the failure wedge is developed throughout the entire ground mass. However, we found that when displacement occurs in the horizontal direction below the footing for the low values of the rock type ($m_i$ = 5 and 12), the results produced by two methods were very similar (Figure 10). In addition, it was also proven that the correlation between results decreases when the horizontal displacement in the area under the footing reduces for the higher values of the rock constant ($m_i$ = and 32).

Similarly, in the vertical direction, when displacements were concentrated in the area attached to the foundation, the correlations obtained were worse between the results of the two calculation methods (Figure 11), so, for example, in Figure 11d the vertical displacements affect very shallow ground depth (great dispersion of results), in contrast to Figure 11a (closed results).

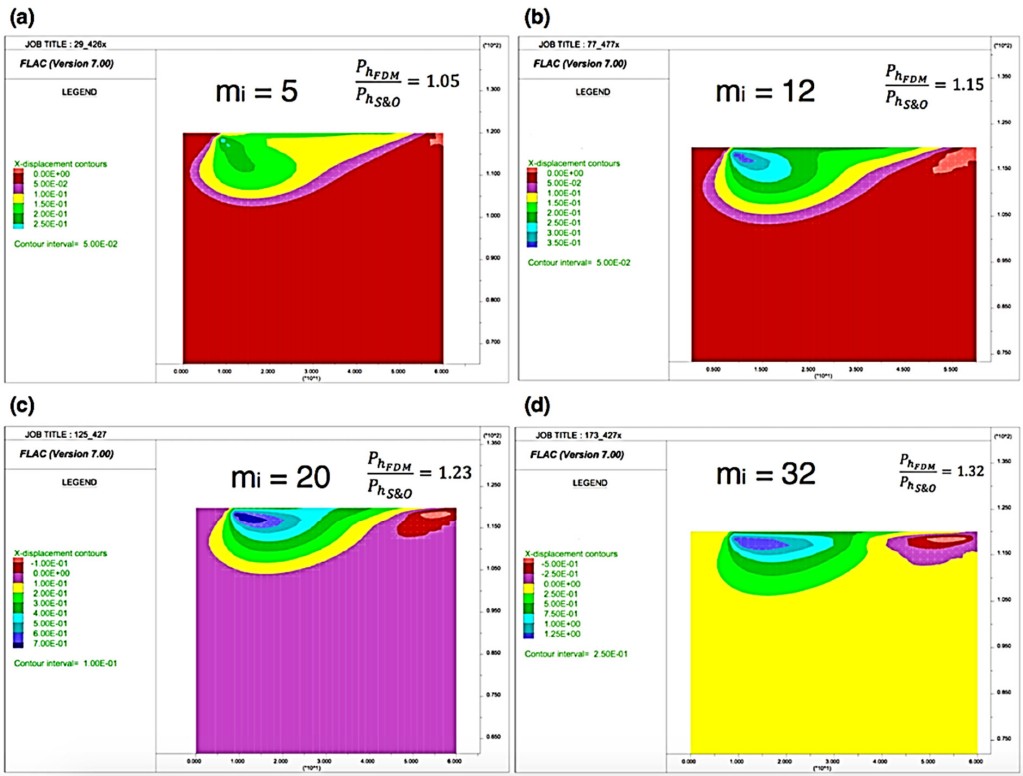

**Figure 10.** The variation of the horizontal displacements under the footing as a function of $m_i$ (B = 16.5 m, *UCS* = 10 MPa and *GSI* = 50). (**a**) $m_i$ = 5; (**b**) $m_i$ = 12; (**c**) $m_i$ = 20; (**d**) $m_i$ = 32 (*x*-axis and *y*-axis units in meters).

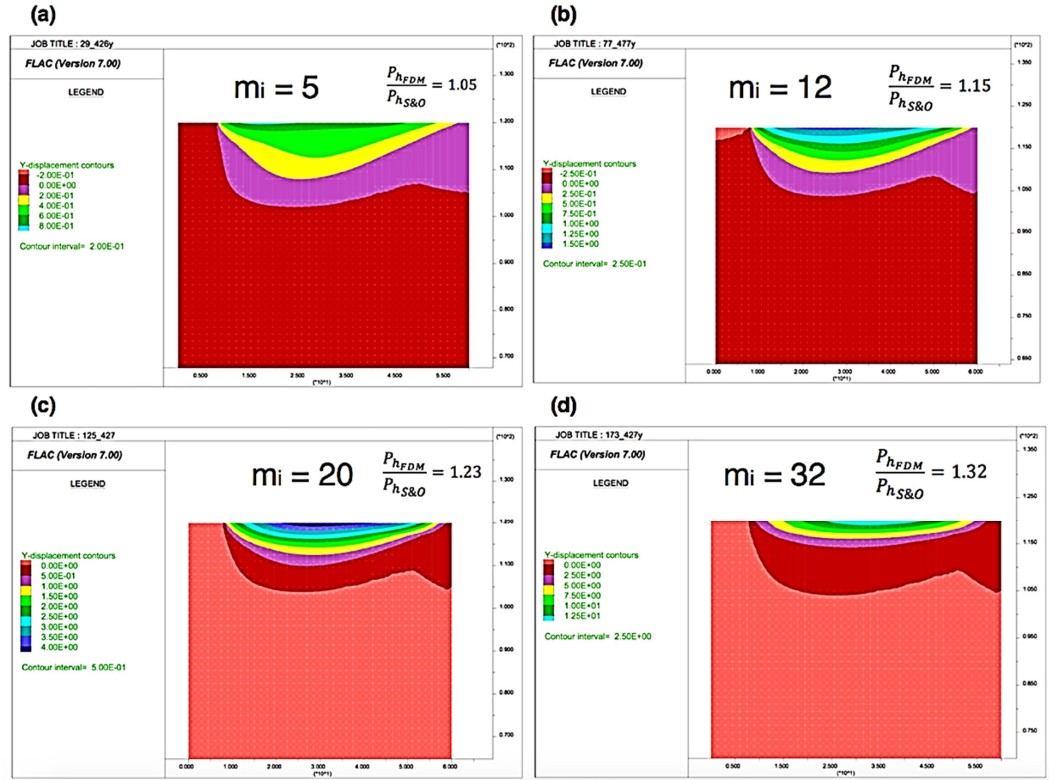

**Figure 11.** The variation of the vertical displacements under the footing as a function of $m_i$ (B = 16.5 m, *UCS* = 10 MPa and *GSI* = 50). (**a**) $m_i$ = 5; (**b**) $m_i$ = 12; (**c**) $m_i$ = 20; (**d**) $m_i$ = 32 (*x*-axis and *y*-axis units in meters).

### 4.2. The Influence of the Self-Weight on Bearing Capacity

Once the difference between the analytical and the numerical results of the bearing capacity is defined as in Section 4.1 for the weightless rock mass, the next step would be to study the influence of the self-weight on the bearing capacity by the numerical method since this analysis allows understanding the influence of foundation width.

With the results of the bearing capacity considering weightless rock mass ($P_{hWL}$) and the bearing capacity with the self-weight ($P_{hSW}$) deduced from the FDM, the correction coefficient was developed due to the self-weight of the rock mass ("weight factor", $W_F$). It should be noted that $P_{hFDM}$ presented in Section 4.1 is defined as $P_{hWL}$ in this section, 4.2, to identify the self-weight cases and follow the abbreviations.

The influence of four variable parameters ($m_i$, *UCS*, *GSI*, and *B*) in the correlation of the results between $P_{hSW}$ and $P_{hWL}$ were also analyzed to develop the correction coefficient.

Figure 12 shows that the four parameters that were analyzed affect the correlation between the results obtained with the hypothesis of the weightless rock mass and considering the self-weight of the material, with the influence of the *GSI* and the *UCS* being the most influential parameters. In Figure 12c,d, a clear decrease in the dispersion range is observed with the increase of *UCS* and *GSI*. It must be emphasized that for *GSI* = 10 the variation between the results reaches almost 400%, while for *GSI* = 50 was close to 80% and for greater values of *GSI* = 85 this variation dropped to less than 20% (Figure 12d).

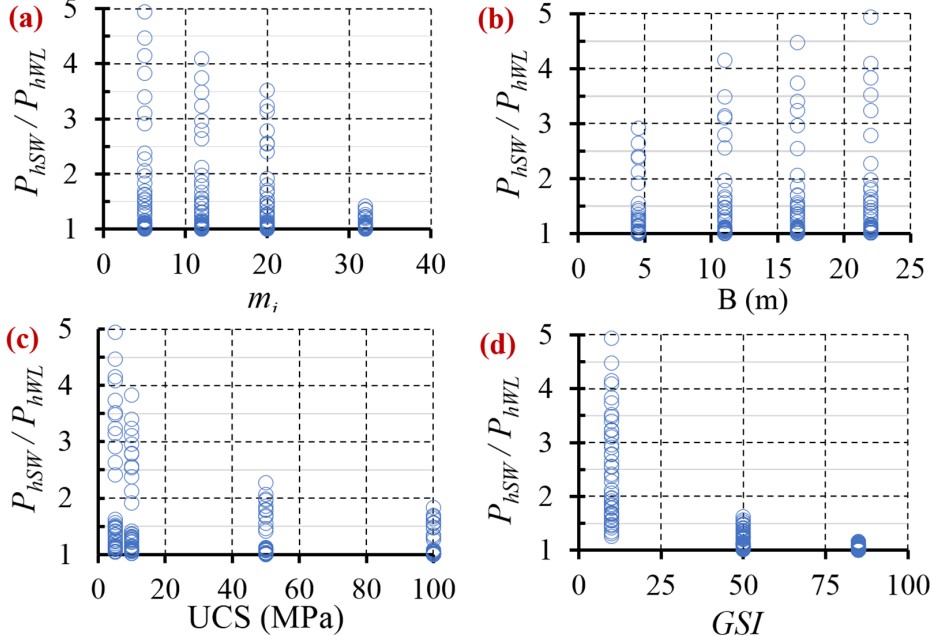

**Figure 12.** Correlation between $P_{hSW}$ and $P_{hWL}$, depending on: (**a**) $m_i$; (**b**) *B*; (**c**) *UCS*; (**d**) *GSI*.

Regarding the influence of $m_i$ (Figure 12a), in general, the tendency is that the higher the value of $m_i$, the smaller the dispersion between results ($P_{hSW}$ and $P_{hWL}$). While Figure 12b shows the increase in the value of dispersion between results ($P_{hSW}$ and $P_{hWL}$) with the increase of the value of *B*, as the foundation width is directly related to self-weight of the rock mass (greater widths of the foundation on the ground imply more rock mass affected, therefore, greater failure wedge and also more influence of its weight). These variations in the results clearly show the need to incorporate the influence of the foundation width in the calculation of the bearing capacity.

From Figure 12, it can be concluded that $m_i$ and *B* affect the dispersion of the results mainly in combination with other parameters, in particular with low values of *GSI*. In cases where the range between $P_{hSW}$ and $P_{hWL}$ exceeded 60% when associated with *GSI* = 10, the dispersion range was very dependent on the values of $m_i$ and *B*.

Regarding *UCS*, it can be observed in Figure 12c that variation was much greater between cases with low *UCS* (5 and 10 MPa), than among those with higher *UCS* (50 and 100 MPa). In other words, as the value of the *UCS* increased, less dispersion was observed, and this decrease occurred exponentially.

Figure 13 shows that the most influential parameter was the *GSI*. For higher *GSI* values (*GSI* = 85), the dispersion was smaller than 10%. However, a lower value of the *GSI* implies a greater influence of other parameters, for example, in cases with *GSI* = 10 all the studied parameters influenced the relation $P_{hSW}/P_{hWL}$.

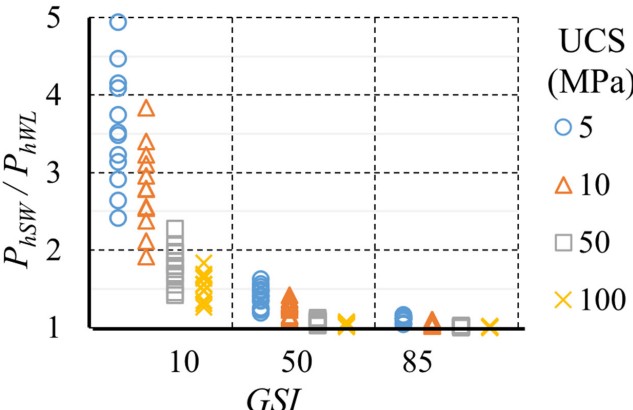

**Figure 13.** Correlation between $P_{hSW}$ and $P_{hWL}$ depending on *GSI* and *UCS*.

In turn, Figure 14 shows that the bearing capacity lower than 2.5 MPa was the most conditioned by the consideration of the self-weight of the material, with the difference in the values at least doubled. For the bearing capacities in the range between 2.5 and 25 MPa the influence of the self-weight was very variable, increasing between 1 and 2 times. Additionally, for the values of the bearing capacity greater than 25 MPa little influence of the self-weight of the material was observed, decreasing the difference for greater values of the bearing capacity; lower or around 20%. It is emphasized that the value of 25 MPa is the average value of the compressive strength of concrete, as well as the limit to define the soft and hard rock according to ISRM [52]. Thus, the cases where the self-weight present more influence on the bearing capacity are the same that the bearing capacity can condition the completion of the project.

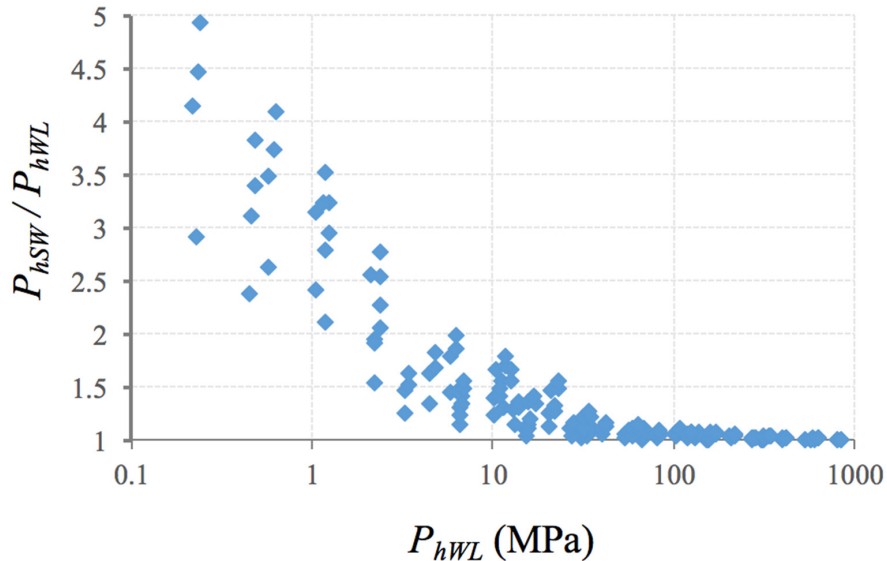

**Figure 14.** Correlation between $P_{hSW}$ and $P_{hWL}$, depending on $P_{hWL}$.

Self-Weight Correction Factor ($W_F$)

The analysis of the results obtained numerically for $P_{h_{SW}}$ and $P_{h_{WL}}$ demonstrated that the correlation of the bearing capacity presents a great dispersion depending on the state of the rock mass (*GSI*), *UCS*, and the footing width (*B*). In addition, the rock type ($m_i$) has very little effect on the correlation of results for $P_{h_{SW}}$ and $P_{h_{WL}}$.

In many cases, notably with the combination of high values of *GSI* and *UCS*, the increase in load due to the consideration of the self-weight of material was less than 5% (Figure 13). Consequently, it is not useful to perform a detailed numerical calculation to estimate the increase in bearing capacity because the self-weight is too small. For this reason, and to allow for a better adjustment of $W_F$, the cases in which the load increase was less than 5% are represented in the graph of the Figure 15.

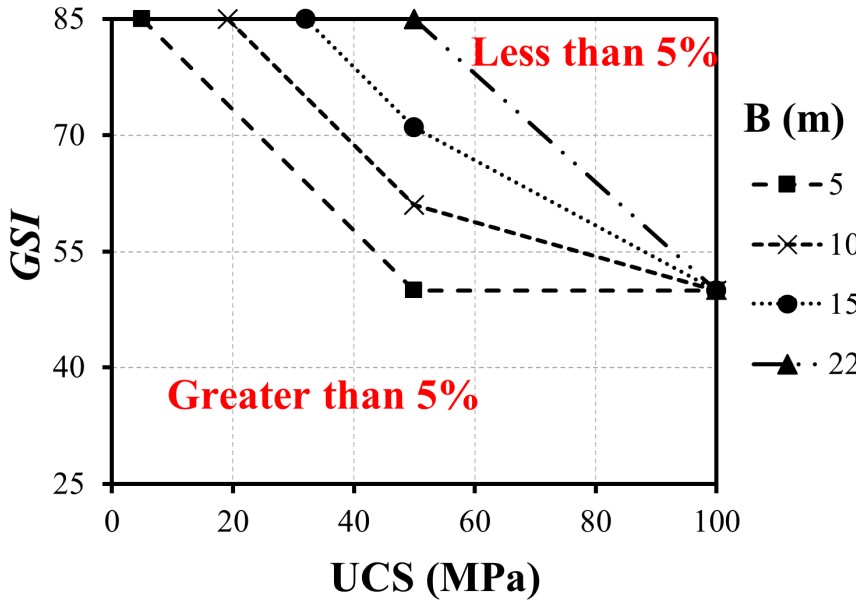

**Figure 15.** Limit of 5% increase in the bearing capacity due to the self-weight of the material.

From Figure 15, knowing the *GSI* and the *UCS* of the rock mass, depending on *B*, we can know whether the consideration of the self-weight of material increases the bearing capacity by more than 5%. This is because the lines for different *B* values delimit the differences smaller than 5% for the combination of *GSI* and *UCS* above each line for each of the *B* values studied.

For example, for a footing width of *B* = 10 m, all combinations of *UCS* and *GSI* that are below the line corresponding to *B* = 10 m show an increase in the bearing capacity, due to the self-weight of material, that exceeds 5%.

If the footing has the width of *B* = 12 m, a line between *B* = 10 m and *B* = 15 m should be interpolated, and the points resulting from the combination of *GSI* and *UCS* that are below the line present an increase in the bearing capacity due to the self-weight of the material greater than 5%.

Once we separated the case studies with an increase lower than 5%, we then developed the correction coefficient taking into account the self-weight ($W_F$). To make the equation dimensionless, it was divided by the lower value of the ultimate bearing capacity correspondent to the case weightless ($P_{h_{WL}}$) and therefore it is proposed in the present research:

$$\frac{P_{h_{SW}}}{P_{h_{WL}}} = 1 + \frac{\Delta P_h}{P_{h_{WL}}} = 1 + W_F \tag{13}$$

Figure 16 represents the correction coefficient ($W_F$) divided by $\sqrt{B}$ versus the *UCS*, obtaining the fitting equations (also shown in Table 4) with high correlation coefficients. In

the graph corresponding to *GSI* = 85 in Figure 16 there are no equivalent columns to the *UCS* of 50 and 100 MPa, because for these combinations of parameters the variation of the bearing capacity with and without the self-weight of the material is less than 5%.

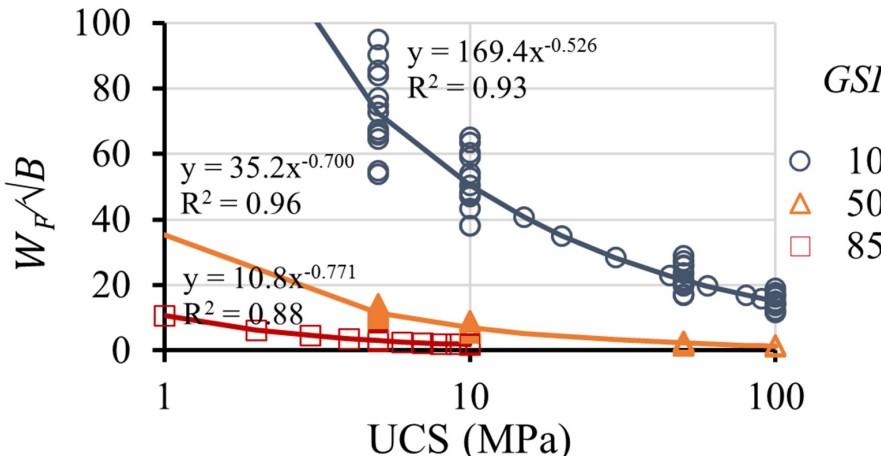

**Figure 16.** $W_F$ graphs based on the *UCS*, for different values of *GSI*: *GSI* = 10; *GSI* = 50; *GSI* = 85.

**Table 4.** $W_F$ equations based on *UCS* for different values of *GSI*.

| GSI | Equations |
|---|---|
| 10 | $W_F = 169.37 \cdot UCS^{-0.526} \cdot \sqrt{B}$ |
| 50 | $W_F = 35.233 \cdot UCS^{-0.7} \cdot \sqrt{B}$ |
| 85 | $W_F = 10.79 \cdot UCS^{-0.771} \cdot \sqrt{B}$ |

Since the equations in Table 4 have the same structure, a single equation can be generalized (14) as a function of the *GSI*, that allows to obtain a new coefficient as a result of this study.

$$W_F = \left( \frac{3000}{GSI^{1.2} \cdot \left( \frac{UCS}{\sigma_{ref}} \right)^{\frac{165 - GSI}{300}}} \right) \cdot \sqrt{\frac{B}{B_{ref}}} \tag{14}$$

$$\sigma_{ref} = 1 \text{ MPa}; \; B_{ref} = 1 \text{ m}$$

Expression (14) allows the consideration of the influence of the foundation width (*B*) in the bearing capacity solution, and clearly states that it depends on the combination of the geomechanical characteristics defined by the *UCS* and *GSI* parameters (as also indicated in the graph of Figure 13). In this way and as indicated, the foundation width is incorporated by the consideration of the self-weight of the ground in the solution, which can be introduced from the coefficient $W_F$ applicable to the weightless solution that is usually obtained in analytical calculations.

From Figure 17 it can be compared, in the 192 cases studied, the numerical results using the correction coefficient ($W_F$) calculated by (14), with a variation which is less than 1% in all calculated cases.

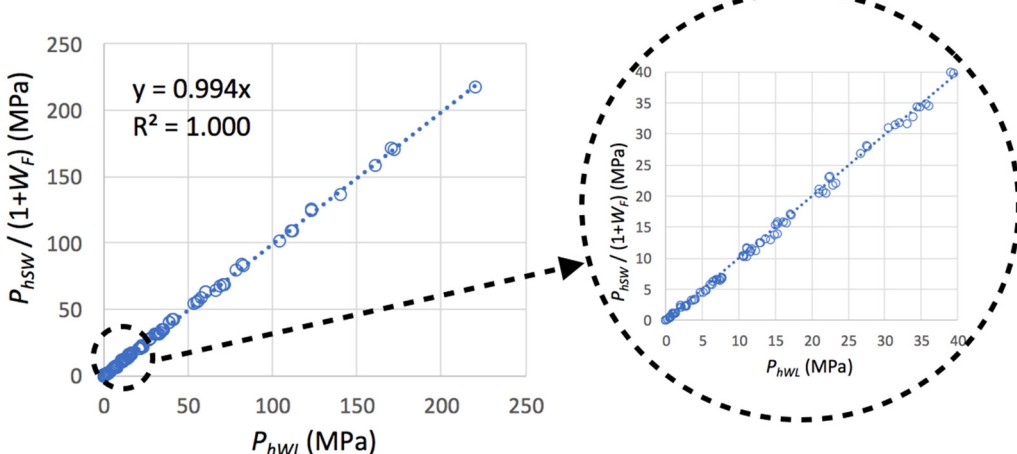

**Figure 17.** Correlation obtained between the numerical results applying the self-weight coefficient Equation (14).

## 5. The Influence of the Self-Weight on Bearing Capacity

In the absence of a coefficient that takes into account the increase in the bearing capacity due to the material's self-weight in rock mechanics, the classical soil mechanics self-weight coefficient $\left(\frac{\gamma \cdot B \cdot N_\gamma}{2}\right)$ was considered.

The coefficient $N_\gamma$ is recommended by Vesic [53]. $N_\gamma$ depends on the value of the friction angle of the material, to apply to a rock mass, and the authors of [54] recommend considering the angle ($\rho_{mean}$) of friction from the harmonic mean of the sine of the instantaneous friction angles acting on the two boundaries as described in Section 2 ($2/sen\rho_{mean} = 1/sen\rho_1 + 1/sen\rho_2$).

To compare the values of bearing capacity obtained using the $W_F$ factor with those factors of the classical soil mechanics equation, the bearing capacity was calculated with the Serrano et al. method [15] ($P_{hS\&O}$). The load increase, due to the self-weight of the rock, was added both by the correction factor presented in this paper (14) ($P_{h1}$) and the classical solution of the soil mechanics ($P_{h2}$).

$P_{h1}$ was estimated by applying $W_F$ to Equation (14), then verified in Figure 15 that for the case of $m_i = 12$, $B = 22$ m, $UCS = 5$ MPa and $GSI = 10, 50,$ and $85$ the self-weight affects the bearing capacity in more than 5%.

$$P_{h1} = (1 + W_F) \cdot P_{hS\&O} \tag{15}$$

$P_{h2}$ is the result obtained using classical soil mechanics [10]:

$$P_{h2} = P_{hS\&O} + \frac{\gamma \cdot B \cdot N_\gamma}{2} \tag{16}$$

Table 5 shows the results of the bearing capacity considering the self-weight of the material through the two methods.

**Table 5.** Bearing capacity considering the self-weight of the material.

| Cases<br>($m_i = 12$, $B = 22$ m, $UCS = 5$ MPa) | GSI | $\rho_1$ (°) | $\rho_2$ (°) | $\rho_{mean}$ (°) | $P_{h1}$ (MPa) | $P_{h2}$ (MPa) | $\varepsilon = \frac{|P_{h2}-P_{h1}|}{P_{h2}}$ |
|---|---|---|---|---|---|---|---|
| 1 | 10 | 64 | 28.8 | 38.8 | 2.2 | 26.2 | 0.92 |
| 2 | 50 | 62.6 | 22.2 | 32.1 | 8.4 | 14.6 | 0.42 |
| 3 | 85 | 53.3 | 19.6 | 28.2 | 30.6 | 30.9 | 0.01 |

The term defined by the bearing capacity considering the self-weight that uses the soil mechanics equation significantly overestimates the bearing capacity. This is due to the fact that every fractured rock mass (low quality) present high friction angles which is the main

influential parameter. In rocks the parameters that qualitatively define and influence the bearing capacity are the *GSI* and the *UCS*, which are used in the coefficient $W_F$; in cases with low *GSI* the influence of $W_F$ is noticeable.

## 6. Conclusions

Existing analytical solutions [15] allow to obtain particular configurations and cannot consider the influence of the foundation dimension. To overcome this limitation, the solutions accepted and widely used in Soil Mechanics contemplate introducing empirical coefficients to improve the real estimate. Similarly, and since there are no such factors in the literature in the field of Rock Mechanics, in this research the correction factor ($W_F$) obtained by numerical experimentation is proposed to offer a complete semi-empirical formulation that considers the dimension of the foundation in the analytical formulation of the bearing capacity of the rock masses.

Based on the comparison of the numerical and analytical results of bearing capacity for conventional foundation widths in bridge construction, the need to consider the influence of the self-weight on the bearing capacity in this type of foundation on rock masses is concluded.

In general, regarding the correction coefficient due to the self-weight ($W_F$) proposed based on the results obtained in 192 cases studied by numerical analysis through FDM, the following can be concluded:

- The parameters that have most impact on the value of the bearing capacity are *GSI* and *UCS*, observing an exponential influence with increasing values of those parameters.
- Depending on the combination of the *GSI*, the *UCS* and the footing width (*B*), the influence of the self-weight of the material may be less than 5% on the value of the bearing capacity in cases with high *UCS* and *GSI* or may exceed as much as 400% for very low values of *GSI* (*GSI* = 10) and *UCS* (*UCS* = 5 MPa).
- Medium or high values of the ultimate bearing capacity ($\geq$25 MPa) are not significantly influenced by the material's self-weight component; the difference being lower than 20%. These case studies correspond to medium or high *GSI* values, as already indicated by Merifiled et al. [7] and Clausen [55].
- The rock type ($m_i$) and the foundation width (*B*) influence the correlation of the results obtained with and without self-weight, however, depending on the combination of the *UCS* and the *GSI*.
- Through the classical soil mechanics self-weight coefficient, the increase in the bearing capacity differs considerably from the estimated using the proposed coefficient for rock masses based on the numerical calculations through the finite difference method. This happens because the rock mass does not have a constant angle of friction, thus depending on the value of the self-weight factor ($W_F$) on *UCS* and *GSI*.
- Based on the numerical and analytical results, the $W_F$ coefficient can be used in conjunction with the analytical method, to estimate in a semi-analytical way the bearing capacity of a bridge foundation, once, due to the foundation size, a great contribution of the self-weight on the bearing capacity is expected.

**Author Contributions:** Conceptualization, A.A.; R.G.; C.O.M. and S.M.; Data curation, A.A. and R.G.; Funding acquisition, R.G. and C.O.M.; Investigation, A.A. and R.G.; Methodology, A.A., R.G. and S.M.; Project administration, A.A. and R.G.; Resources, A.A.; Software, A.A.; Supervision, R.G. and C.O.M.; Validation, A.A. and R.G.; Visualization, A.A.; Writing—original draft, A.A.; Writing— review & editing, R.G. and C.O.M. All authors have read and agreed to the published version of the manuscript.

**Funding:** This research was funded by Universidad Politécnica de Madrid.

**Institutional Review Board Statement:** Not applicable.

**Informed Consent Statement:** Not applicable.

**Data Availability Statement:** Not applicable.

**Acknowledgments:** The research described in this paper was financially supported by the Universidad Politécnica de Madrid from the grant with reference VMENTORUPM21RAGA of the university's own program to carry out research and innovation projects.

**Conflicts of Interest:** The authors declare no conflict of interest.

## Abbreviations

| | |
|---|---|
| $m_i$ | geological origin of the rock mass |
| $\sigma_c = UCS$ | uniaxial compressive strength |
| $GSI$ | geological strength index |
| $\sigma_1$ | major principal stress ($\sigma_1$) |
| $\sigma_3$ | minor principal stress ($\sigma_3$) |
| D | alteration factor |
| m, s | Hoek–Brown's parameter |
| $\alpha$ | inclination of free boundary |
| $f_1$ | load acting on a free surface |
| $i_1$ | inclination of the load on the free boundary |
| $P_h$ | bearing capacity of the foundation |
| $i_2$ | inclination of the load on the foundation boundary |
| $I_a$ | Riemann's invariant |
| $\rho_2$ | instantaneous friction angle at the boundary 2 |
| $\rho_1$ | instantaneous friction angle at the boundary 1 |
| $\Psi_1$ | the direction of the principal stress at the boundary 1 |
| $\Psi_2$ | the direction of the principal stress at the boundary 2 |
| $\beta_a$ | normalized characteristic strength |
| $>\zeta_a$ | tenacity coefficient |
| $N_\beta$ | bearing capacity factor |
| B | foundation width |
| $P_{hFDM}$ | numerical bearing capacity using FDM |
| $P_{hS\&O}$ | analytical bearing capacity |
| $\Delta P_h$ | increment of the bearing capacity observed in numerical method using FDM |
| $P_{hWL}$ | bearing capacity considering weightless rock mass |
| $P_{hSW}$ | bearing capacity with the self-weight deduced from the FDM |
| $W_F$ | self-weight correction factor |
| $\rho_{mean}$ | mean friction of the two boundaries |
| $\gamma$ | specific weight of the ground |
| $N_\gamma$ | bearing capacity factor corresponding to the self-weight in formulations of the soils |

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
