# Peer review of "Assessment of the Bearing Capacity of Bridge Foundation on Rock Masses"

_applsci, doi:10.3390/app112412068_

Round 1

Reviewer 1 Report

In the numerical analysis it was assumed that the bearing capacity was reached when 192 the continuous medium did not stand more load, because an internal failure mechanism 193 had formed. The load in FLAC was applied through velocity increments, and the bearing 194 capacity was know from the relation between stresses and displacements of one of the 195 nodes: for this study we considered the central node of the foundation.

Author Response

The answers are included in an attached file

Reviewer 2 Report

The manuscript is overall interesting. However, some issues are found throughout the paper. Therefore, according to this Reviewer, a moderate revision would be necessary before the paper can be further considered for possible publication in Applied Sciences. All details are summed up in the following.

Required changes:

  1. Despite understandable, English needs some improvements.
  2. Originality/novelty of the study proposed. This issue is very important and should be better clarified and well highlighted in the text.
  3. Introduction: for the sake of completeness, this section could be integrated by adding some literature review about methods for the estimation also of pile foundations (Achmus and Thieken, 2010; Conte et al., 2021).
  4. Generally throughout the whole paper: it would be better to write ‘Hoek-Brown failure criterion’, rather than ‘Hoek and Brown failure criterion’.
  5. Line 112: add the name of the authors before reference 42-43. The same is for 44-45 at line 114.
  6. Equation 1: add the reference that this equation is taken from.
  7. Line 143 and figure 1: from the reading of the text at line 143 and by looking at the figure, it is not clear what i1 is. The combination of the text and figure is somehow confusing. Besides, how is the inclination of the load f1 taken into account?
  8. Line 179: The Modified Hoek-Brown constitutive model was employed. What are the values of the constitutive model parameters?

REFERENCES

Achmus, M., and K. Thieken. 2010. “On the behavior of piles in non-cohesive soil under combined horizontal and vertical loading.” Acta Geotech. 5 (3): 199–210. https://doi.org/10.1007/s11440-010-0124-1.

Conte, E.; Pugliese, L.; Troncone, A.; Vena, M. 2021. "A simple approach for evaluating the bearing capacity of piles subjected to inclined loads." International Journal of Geomechanics (ASCE), 21(11): 04021224. DOI: 10.1061/(ASCE)GM.1943-5622.0002215.

Graine, N., M. Hjiaj, and K. Krabbenhoft. 2021. “3D failure envelope of a rigid pile embedded in a cohesive soil using finite element limit analysis.” Int. J. Numer. Anal. Methods Geomech. 45 (2): 265–290. https://doi.org/10.1002/nag.3152.

Author Response

(The authors gave the same response as above.)

Reviewer 3 Report

  1. Introduction: It is inappropriate to directly quote a large number of references. Each cited reference should be discussed separately and demonstrate its importance to your work. Please describe their claims in one or two sentences and the evidence they provide in support of their claims.
  2. In introduction, many references are cited but without reviewing their limits and benefits.
  3. This paper involves many formulas. It should be clear which are for reference and which are derived by the author. Relevant references should be cited.
  4. This paper has many symbols and abbreviations. It is suggested to add a list of symbols before the introduction.
  5. It is suggested that the author add the innovative description of this paper in the introduction and conclusion.
  6. The discussion section is very descriptive. There needs to be a more appropriate analysis of the data. Moreover, the observations need to be adequately justified.
  7. Improve the conclusions and stress the novelties of the paper.

Author Response

(The authors gave the same response as above.)

Reviewer 4 Report

The author presented a well-written manuscript about a numerical study regarding the influences of foundation dimension and geotechnical parameters on the bearing capacity of foundation in rock masses. Topic falls into the scope of the journal and the conclusions provide useful new information on the topic. I appreciate this manuscript because every figure is authentically analyzed, and the analysis can well support the conclusions drawn at the end of the manuscript. I tend to recommend it publish in the Journal after certain modifications.

Specific comments:

  • Proof reading is needed in sentences: 112-116, 489-490.
  • Line 126: “validate” -> “validated”.
  • Lines 165-170: some symbols are different from those in the equations.
  • Line 171: “ … the ultimate bearing capacity (Ph) is estimated:”, the following equation expresses the bearing capacity factor rather than the ultimate bearing capacity. Pls rewrite it.
  • The size of numerical model shown in Fig. 2 and the unit of horizontal and vertical displacements shown in Figs. 10 and 11 are not given.
  • Line 195: “know” -> “known”.
  • The figure number of “Correlation of PhFDM and PhS&O in function of: (a) mi; (b) B; (c) UCS; (d) GSI” is wrong.
  • The average value should be added in Figs. 4 and 12 to better interpret the variation trend of results.
  • Line 242: a dangling modifier: “This is due because …”.
  • (12) is absent.
  • In Fig. 9, the y-axis title is with dimensionless not with the unit of MPa, so it should be checked carefully.
  • 15 is absent.
  • Sections 4.1.2 and 4.2.1 have identical titles (i.e., Displacement analysis).
  • Only one subsection exists in Section2 and Section 5, thus is it better to delete the subsection?

Author Response

(The authors gave the same response as above.)

Round 2

Reviewer 1 Report

Dear authors,

The reviewer apologises: his comments have not been uploaded at the first review: hence this strange comment on the authors’ response. The reviewer, therefore, updated his previous comments according to what has changed in the paper.

The paper presents a very interesting study about soil bearing capacity. Though the topic is bridge foundations, the developments are, as the reviewer understands, applicable to every foundation problem. The authors carried out numerical simulations on case studies and compared the results with analytical predictions. Their aim was to define analytical expressions that depend on the geotechnical parameters to predict (by fitting), the difference between numerical and analytical results.

Though it could be made clearer, the objective is fair, and the designed research seems appropriate. However, a central problem is the accuracy of the numerical models. The boundary on the right of the model does not seem correct from the simulation results. In addition, the application of the force to the ground directly seems strange (homogeneous distribution of forces + fixity of the gridpoints). It leads to a lack of confidence in the numerical results, which are the basis of the paper. Validation of the methodology with respect to experimental results would be welcome

Then, the statistical analysis of the results is not convincing. The authors should build a correlation matrix between the inputs (probably also including PhS&O) and the output (PhWL/PhS&O). They should inspire themselves from similar studies.

Finally, the quality of writing could still be improved to increase the easiness to read the paper. It concerns not only the parts highlighted by the reviewer. The reader is often lost in the paper, and this should be avoided by constructing a clear outline of each part. Please then have careful proofreading, ideally with a native speaker.

In addition to the general comments above, the authors can find below major, minor comments as well as typos.

Major comments:

In general, the reader is lost in the introduction. The division analytical/numerical works is fair. However, each part should be better organised and follow a path to guide the reader to the final outcome: e.g. "what's globally missing in the analytical method?" "What is possible to study through numerical methods?" Though some parts are collections of small descriptions of different contributions, a final comment should always sum up the contributions clearly to show what's still missing and what should be investigated. It's really helping to understand the contribution of the paper.

Though the second section gives valuable insights into the analytical model considered in the paper, the reader does not really understand the point of showing this here. Do the authors only explain the model of Serrano et al [15]? If so, the authors could go more straight to the point and be more pedagogic to allow the reader to understand what is really important for that paper and what's not. Otherwise, please clearly state what is new here. In any case, the reviewer suggests rewriting section 2 to make clearer the point of the authors, which is not clear now.

Lines 222-229 are really fuzzy. The analysis carried out is not clear at all, and the reasons even less. Please, first state the general objective of the section. Then, define the model the authors used and why, including the basic hypotheses, e.g. plane strain, i.e. the one that will not change during all the paper. Finally, present the different steps you follow and the reason for each of them (if relevant, but probably not). Comment lines 235-237 should, for example, be included in the main hypotheses. Does this last comment mean that the load is always homogeneously distributed on the ground surface, irrespective of the ground deformation? If so, it is a strange choice because it is assuming that the foundation is highly deformable, which is far from reality. Also, Figure 2 shows that the gridpoints modelling the foundation are all fixed. It is again strange because one would assume some deformation, especially there.

Line 242-246: the paragraph is not clear. Either this is classical and do not need much explanation, or it is not classical and needs to be explained better. Also, it would be better to present the varied geotechnical parameters first and then the results: it is more intuitive. Additionally, the reviewers asks how the stress and displacement at the central node of the foundation can vary through the time. The gridpoint is fixed => no displacement.

Figure 2: the model seems a bit small in length, especially for the foundation width B=22m who reach more than half of the model, which is too big, one third would have been better. Could the authors validate with graphs showing no deformation close to the boundaries for some cases with B=22m? As seen in Figures 10 and 11, the length of the model depended on the width of the foundation. In the reviewer's opinion, it leads to a less relevant comparison between cases. In addition, both Figures show that some stresses are present close to the boundary, which means that the boundary is either too close or too appropriately chosen. Probably, the depth of the model is too big, though. This too-close boundary condition might lead to a more "rigid behaviour" of the soil at the boundary, leading in the end to a higher bearing capacity. Therefore, the trend/dispersion obtained by the authors could be linked to this numerical issue.

To the reviewer opinion, Table 2 only shows again what has already been seen from Figure 4a and Figure 4c. mi influences PhFDM/PhS&O but not UCS. Paragraph lines 316-322 should either be deleted or be condensed, only stating that Ph is directly related to UCS. If so, a simplified equation should be given. Indeed, this dependency is already known, so nothing new has been found here, in the reviewer's opinion.

The aim of Figure 6 is not clear at all. What do the authors want to say with this graph? The comments line 328-329 is not completely true. The authors do not really know if what plays a role is the higher Ph or the GSI. In general, the authors identified a clear dependency between PhFDM/PhS&O with mi. The reviewer thinks they should then normalise by this found trend to see if the other parameters (UCS, GSI, B) explain the dispersion obtained in Figure 4a. For now, all these graphs are fuzzy. Section 4.1 (before 4.1.1) should be highly condensed.

The further scatter observed in Fig 8 could probably be explained by the foundation width. Since the authors mentioned it in Figure 5, they should justify why it is not present in the final correlation equations.

Figure 9 scatter does not look correct to the reviewer, given the observed scatter in Figure 8. For instance, where did the highest point (GSI = 50 and mi = 32) of Figure 8 go on Figure 9. The reviewer does not believe it leads to a smaller scatter than 4%.

Given the values obtained in Figure 12 (in average more than 2), how do the authors justify studying the WeightLess case? Why the authors did not do directly the correlation of PhS&O with PhSW?

The comparison presented in Section 5.1 should also comprise a "real" simulation to compare with measured values from experiments, not using the analytical formulas derived from the paper. The authors would even show better the potential of the correlation factors.

Minor comments:

Line 17: better than "rock mass type", the authors could directly say "rock mass weight" to be clearer0

Line 19: consider adding "classical" in "adopting the classical hypotheses of plane", not mandatory

Line 20-23: The sentence is not clear. Is it analytical? Numerical? If it's numerical, please remove it since this is already said a few lines before. If it is analytical, please only say just before something like: "associated flow rule; the rock mass weight being variable" or "associated flow rule and variable rock mass weight". This should be clarified in the abstract.

Line 23: "Thus, a self-weight correction factor is proposed" can be removed and integrated into the next sentence: "analytical results, a correlation factor function of the self-weight is proposed. It can be used in conjunction"

Line 13-15: The reviewer would rather read first, "However, its effect has been shown to be very significant in soils with linear failure criteria (Mohr-Coulomb envelope), and its study is necessary in the case of non-linear failure criteria, typical of rock masses" than "Analytical solutions do not allow incorporating the dimension of the foundation." First, the authors state what they want to study and why (which is clear), and then how and why, which is also clear

Line 63: "the numerical method", "the" seems strange to the reviewer + could it be more precise. What kind of numerical method is it?

Introduction: Line 65-67 => Mohr-Coulomb model // Line 67-68: Hoek and Brown model // Line 68-71: Mohr-Coulomb model // Line 71-73: Hoek and Brown model. The reader does not understand why the authors mixed the two models. Either make clearer the link (if relevant) or better separate the two models talking first about MC and then about H-B.

Line 72-73 are not well written. It is better to state first the difference between H-B and MC (non-linear failure criterium). And then, the authors can add the fact that similar to MC, H-B is isotropic, thus unable to model rock discontinuities properly.

Line 150-151: "however, they do not claim to be general for consideration in the presentation of results" What does it mean?

Line 184: the factor mi should be specified more clearly. Where does the equations 2, 3, and 4 come from?

Figure 1: the angle alpha and i1 seem similar. Are they equal? If so, please use only one parameter for both.

Equation 5: the value Phi1 and Phi2 are not defined in the text.

Line 215-216 basically repeats what's said lines 199-203.

Equations 7 & 8 directly come from Equations 5 and 6? This should be stated. Similarly, where does equation 9 come from? Also, it seems that "sin" has been misspelt into "sen" in Eq. 9

The first sentences of section 3 should be better written. The authors can recall the objectives, which is to "numerically model accurately real bridges foundations and to compare to the described analytical solution." "To do so, numerical simulations have been run using FLAC software (a reference to ITASCA would be welcome), which uses an explicit finite difference (FDM) formulation".

Line 267-271 repeats partially what is stated lines 226-229. Why are the first 2 UCS values so close one to another compared to the other two differences? Maybe a plot in logx would be better with the chosen values.

Line 284-286: did the authors try to plot a correlation matrix of this collection of results to better support their statement.

Figure 4b does not show any influence, in the reviewer's opinion. From Figure 5, it seems that B=16 and B=22 give the same results and that B=4.5 and B=11.5 also give the same. What is strange is that B=4.5 always seems higher than B=11.5 (not a linear trend, then). This result would be better stressed with a correlation matrix probably.

Sentence line 303-306 is not clear.

In table 2, it seems that many cases give PhFDM/PhS&O < 1. Are there any mistakes?

Figure 6: please always use GSI and not move to RMR in the figure.

Figure 8 y abscise should precise that the values are percentages.

Figure 9 seems to miss something in the equation: PhFDM/(1+DPh/PhS&O). Please also increase the quality

Line 417-418: please use consistently the same variables to describe the same things all along with the paper.

Line 535: The correlation factor should also consider Equations of Table 3 to be correct.

Some typos:

The email address of the corresponding author should be given not only the mail address

The first sentence "The aim of this paper was to study" => "This paper aims to study".

Author Response

The answers are attached in the attached file

Reviewer 2 Report

The replies provided by the authors are sufficient to address the issues previously raised. Accordingly, the manuscript can be accpeted for publication in the present form.

Author Response

Thanks

Reviewer 3 Report

The author made corresponding improvements according to the comments of the reviewers. However, the reviewer still feels that further improvement should be made. The conclusion is too long. It is suggested to reduce and refine the conclusion.

Author Response

Ok, thanks to the reviewer. Conclusions are simplified for a better understanding.

Reviewer 4 Report

The authors basically addressed my comments except that the y-axis title in Fig. 9 should be “PhFDM/(1+ ∆Ph/PhS&O)”. Since ∆Ph is with the unit of pressure in this manuscript, from dimensional analysis x-axis and y-axis would not be equal as that in the figure.

Author Response

Ok. Thanks. The reviewer is right, it has been corrected in figure 9.

Round 3

Reviewer 1 Report

Dear authors,

As a follow-up, the reviewer will react on the previous points.

In general, as already stated, the aim is interesting, but the paper is still not publishable in the present form. Details about the numerical method are missing (stiffness, way to apply the load). The performance of the model does not seem consistent. The reviewer asks for a “real” deep revision of the paper, including the outline of the numerical section.

Point 1: Regarding the boundaries of the model, why the authors used a so big depth with respect to the width? Did they always notice the strange bulb (top-right corner) for wider models? How the authors explain the fact that though the external gridpoints are fixed (cf Figure 2), it seems that there is some modification at the top-right of the each model? On the right side of the model, is a boundary condition with fix X and free Y possible? Is it better, in the authors’ opinion?
In Figure 11, do the authors think that the wedge is far enough for the right boundary?
In Figure 11b, how the authors explain that at the load application location there is negative Y-displacement?
Regarding the way the load is modelled? Is it more complicated to model the foundation through a rigid block with attached gridpoints to the soil? Is it more accurate?

Point 2: In the reviewer’s opinion, there are too still many typos and not well-written sentences. How many hours did the native speaker spend on the paper?

Point 3: The repetition of “ but they do not vary the foundation dimensions” is a bit exaggerated to the reviewer’s opinion (Line 106-116). In addition, the reviewer does not find clear if it means that the authors did not study the width influence / could not study the width influence or showed that the width did not have any influence.

The introduction is still very fuzzy to the reviewer’s opinion and has not been deeply reworked. The distinction between analytical (i.e. theoretical) FE limit analysis and FE simulations as carried out throughout the paper are not clear. Do the present simulations (section 3) use a classical FE approach with material stiffness? What is the stiffness parameter used for the soil?

Point 4: Though the reviewer agrees on the importance of showing the final closed-form equation of Serano et al [15], he thinks that lines 200 – 220 could be reduced to only Eqs 7,8 & 9. Indeed, all the text and the other equations are not self-contained (with respect to ref 15). The reader can not understand the equations, and moreover some terms are not defined (p1, p2).

Point 5: The way the load is applied is now clear from the authors’ answer. However, it should be clarified in the paper, in the reviewer’s opinion. Do the authors think that adding another legend for the nodes where the load is applied could make this much clearer?

The reviewer agrees that deformation is inexistent in analytical formulations. That’s why the authors used numerical simulation, isn’t it? To have a more realistic problem. So the soil deformation matters.

Point 7: see Point 1. The reviewer talks about displacement since it’s what is shown on Figure 10 and 11.

Point 9: The reviewer still finds this Figure hard to understand. How do the authors use the given information (Thus, it can be concluded that a higher value of the bearing capacity implies a lower dispersion of results between both solutions.) afterwards?

Point 10: The reviewer does not agree. The top triangle of Figure 8 is approximately at dPh / PhS&O = 55% (the reviewer guesses these are percentages), while the regression gives 35%.  It means that PhFDM, simulation = PhS&O * 1.55. If dPh is estimated through the regression, we have 1+dPh/PhS&O = 1.35. Therefore Ph FDM /(1+dPh/PhS&O) = 1.55/1.35 = 1.15 which is difficult to fit the maximum scatter of 4% indicated by the authors. Note that the y-legend seems to miss something PhFDM/(1 + dPh/PhS&O).

Point 11: The reviewer was suggesting to remove the equations of table 3 and directly show equation 13 relating PhSW to PhS&O.

Point 12: Do the authors think that they could at least validate the numerical mode on one of these small-scale experiments?

In addition to the general comments above, the authors can find other minor comments as well as typos.

Minor comments:

Line 80: It is first introduced here. Does the foundation width B refer to X-direction of Figure 2? Probably a reference to the Figure 1 (with the appropriate legend) could help the reader to understand. The equivalence between weightless rock and no influence of foundation width can be also better explained, unless very classical in the field. Finally, why does Figure 1 present a finitely wide foundation if analytical formulations can not account for the foundation width?

Line 103: Does the author mean “depth” (z direction) or “width” (x-direction)?

Line 106-108: why these refs (26, 27) are presented in numerical methods since they are analytical/theoretical? Do the authors mean that the proposed approaches can not account for the foundation width? Or that they showed that the width did not have any influence?

Line 111: “where the results are independent of the size foundation” Do the authors mean that the proposed approaches can not account for the foundation width? Or that they showed that the width did not have any influence?

Figure 16 seems to show simulations with values for UCS different than the one written in Table 1 (5 10 50 100). Why?

Some typos:

Line 108: an theoretical => a theoretical

Line 109: no depending of the width => do not depend on the width

Line 116: an particular => a particular

Line 131: although if the GSI≤10 this method is ???

Line 170: offer => offers

Line 200: need => needs

Author Response

The typo are corrected.

Minor comments are clarified in the manuscript: (1) the foundation width corresponds to the area in contact with the ground. The analytical solution gives the same result regardless of width, because loading ends where the analytical model contour ends at contour 2 (regardless of width); (2) obviously the width corresponds to the area in contact with the ground and the depth indicates the foundation elevation; (3) References 26 and 27 add solutions to the analytical formulation (corrected in text); (4) the analytical solution can not account for the foundation width; (5) only values ​​of 5, 10, 50 and 100 are used to represent Fig. 16.

Regarding major changes: (1) The indicated effects of the contour have been verified as said and are two orders of magnitude lower as indicated by the legend of the graphic outputs. In addition, convergence studies have been carried out in this regard; (2) typos are corrected; (3) as indicated by the reviewer, Fig. 4b does not mean the influence of the foundation width; (4) The analytical formulation has been summarized with the essential equations, outside of which the formulation cannot be applied. For example, if the Riemann invariants are not presented, the reader can hardly apply equations 7, 8 and 9; (5) The application of the load is indicated in the manuscript, fixing the nodes and applying a velocity to infer a force; (9) Figure is dicussed in the text ; (10) For the statistical analysis, the reviewer has gone to the worst case fit, which has a deviation of 15%, however with respect to all cases the global error is only 4%; (11) We think that Table 3 provides relevant information to the manuscript; (12) On a small scale, the width of the foundation has no influence, obviously, so the solution is the analytical one that has already been validated in articles on the matter. 

Reviewer 3 Report

Accept in present form

Author Response

Thanks